# In Vivo Modelling of Hepatitis B Virus Subgenotype A1 Replication Using Adeno-Associated Viral Vectors

**DOI:** 10.3390/v13112247

**Published:** 2021-11-09

**Authors:** Shonisani Wendy Limani, Njabulo Mnyandu, Abdullah Ely, Reubina Wadee, Anna Kramvis, Patrick Arbuthnot, Mohube Betty Maepa

**Affiliations:** 1Wits/SAMRC Antiviral Gene Therapy Research Unit, Health Sciences Faculty, School of Pathology, University of the Witwatersrand, Private Bag 3, WITS 2050, Johannesburg 2000, South Africa; wendylimani@gmail.com (S.W.L.); njabulo.mnyandu1@students.wits.ac.za (N.M.); abdullah.ely@wits.ac.za (A.E.); Patrick.arbuthnot@wits.ac.za (P.A.); 2Department of Anatomical Pathology, Health Sciences Faculty, School of Pathology, University of the Witwatersrand, Private Bag 3, WITS 2050 and National Health Laboratory Services, P.O. Box 1038, Johannesburg 2000, South Africa; reubina.wadee@nhls.ac.za; 3Hepatitis Virus Diversity Research Unit, Health Sciences Faculty, School of Clinical Medicine, University of the Witwatersrand, Private Bag 3, WITS 2050, Johannesburg 2000, South Africa; anna.kramvis@wits.ac.za

**Keywords:** adeno-associated virus, anti-viral gene therapy, hepatitis B virus, HBV mouse models

## Abstract

The paucity of animal models that simulate the replication of the hepatitis B virus (HBV) is an impediment to advancing new anti-viral treatments. The work reported here employed recombinant adeno-associated viruses (AAVs) to model HBV subgenotype A1 and subgenotype D3 replication in vitro and in vivo. Infection with subgenotype A1 is endemic to parts of sub-Saharan Africa, and it is associated with a high risk of hepatocellular carcinoma. Recombinant AAV serotype 2 (AAV2) and 8 (AAV8) vectors bearing greater-than-genome-length sequences of HBV DNA from subgenotype A1 and D3, were produced. Transduced liver-derived cultured cells produced HBV surface antigen and core antigen. Administration of AAV8 carrying HBV subgenotype A1 genome (AAV8-A1) to mice resulted in the sustained production of HBV replication markers over a six-month period, without elevated inflammatory cytokines, expression of interferon response genes or alanine transaminase activity. Markers of replication were generally higher in animals treated with subgenotype D3 genome-bearing AAVs than in those receiving the subgenotype A1-genome-bearing vectors. To validate the use of the AAV8-A1 murine model for anti-HBV drug development, the efficacy of anti-HBV artificial primary-microRNAs was assessed. Significant silencing of HBV markers was observed over a 6-month period after administering AAVs. These data indicate that AAVs conveniently and safely recapitulate the replication of different HBV subgenotypes, and the vectors may be used to assess antivirals’ potency.

## 1. Introduction

Hepatitis B virus (HBV) is a small, enveloped virus of the family *Hepadnaviridae*. Infection of the liver by HBV may cause acute or chronic infection, and chronic carriers are at a high risk for cirrhosis and hepatocellular carcinoma (HCC) [1,2,3]. Globally, two billion people have been infected with HBV, and 296 million are currently chronically infected. About 820,000 people die each year as a result of HBV-related cirrhosis and HCC [4]. HBV is classified into at least nine genotypes, A to I, and except for E and G, all other genotypes are further subdivided into subgenotypes [5]. The genotypes, and in some instances subgenotypes, have distinct geographical distributions, manifesting in different clinical consequences and responses to therapy. In South Africa, where approximately two and a half million people are chronically infected with HBV [6], subgenotype A1 prevails [7]. This subgenotype is associated with a particularly high risk for HCC [8]. Subgenotype A1 has distinct molecular characteristics differentiating it from subgenotype A2, which prevails outside Africa [9]. Several HBV subgenotypes exist with different disease manifestations, responses to therapy and geographical distributions [5,10].

There are two classes of licensed anti-HBV drugs on the market: (1) interferon-alpha (IFN-α) and its pegylated derivatives, and (2) nucleotide/nucleoside analogues that inhibit viral reverse transcriptase (RT) [11,12]. However, the effectiveness of these drugs is limited, and functional cure with available treatment is uncommon. IFN-α may cause adverse side effects after long-term use and RT inhibitors may lead to the selection of escape mutations during error-prone HBV DNA replication [13]. The use of gene therapy to disable HBV replication has shown promise and may be well suited to eliminating covalently closed circular (cccDNA) [14,15]. Anti-HBV gene therapies under investigation include silencing strategies that harness RNA interference (RNAi) and gene editing with engineered transcription activator-like effector nucleases (TALENs) and the clustered regularly interspaced short palindromic repeats (CRISPR)/CRISPR-associated (Cas).

Given the large number of individuals chronically infected with HBV, the seriousness of complications and variability in clinical course associated with different subgenotypes, developing tailored novel therapies for HBV infection is a global priority [16]. However, clinical translation requires evaluation in suitable models of HBV replication. These models are also important to elucidate virus–host interplay, mechanisms of persistence and pathogenesis. Apart from humans, chimpanzees are the only animals that are fully susceptible to HBV infection. However, several limitations, including large size, high cost and ethical concerns, restrict their use. To render macaques permissive to HBV infection, human sodium taurocholate co-transporting polypeptide (hNTCP)-encoding sequences were delivered to hepatocytes of these primates [17]. This approach has not yet been widely used but shows promise. HBV-transgenic mice have been very useful in efforts to elucidate the mechanisms of the viral pathogenesis [18]. However, HBV does not naturally infect murine hepatocytes, and formation of cccDNA is not supported in murine hepatocytes. Viral entry, spread and the complete replication cycle are therefore not simulated in mice. Furthermore, variability in HBV gene expression, immunological tolerance to viral antigens and high levels of HBV replication are complications that limit the utility of HBV-transgenic mice.

As alternatives to transgenic mice, adenoviral (Ad) and adeno-associated viral (AAV) vectors have been used to deliver HBV genomes and initiate HBV replication in mice [19,20,21]. AAVs are generally favoured vectors because their genomes are easily manipulated, their immunogenicity is attenuated, they are capable of the efficient transduction of a wide range of cell types, and they achieve long-term transgene expression in transduced cells. AAVs can deliver replication-competent, greater-than-genome-length HBV genotype D sequences to the livers of mice [22,23,24], resulting in persistent HBV replication and the development of liver fibrosis and HCC with minimal acute inflammatory responses. Given that the variation in disease pathogenesis and treatment responses are influenced to a large extent by HBV subgenotypes, modelling subgenotype-specific HBV replication is important [25].

The use of AAVs to model HBV subgenotype A1 is thus valuable in efforts to advance better anti-HBV therapy, and is particularly relevant to sub-Saharan Africa, where the infection is endemic. Here we report on use of AAVs to model HBV A1 replication in cultured cells and in vivo. This model supports the long-term replication of HBV, which mimics the natural chronic infection. The minimal immune responses to AAV vector, together with stable replication markers, are advantages over transgenic mice and Ad-delivered HBV genomes. Moreover, we show that the model is useful to evaluate expressed anti-HBV gene silencers against subgenotype A1.

## 2. Materials and Methods

### 2.1. Construction and Characterisation of AAV Plasmids Containing Greater-Than-Genome-Length HBV Sequences

The plasmids containing greater-than-genome-length HBV sequences of subgenotype A1 (pCDNA-1.28 × A1, Accession number KM519452) and D3 (pCDNA-1.28 × D3, Accession number KM519455.1) sequences have been previously described [26]. To allow insertion of the HBV sequences, the cytomegalovirus major immediate-early enhancer/chicken beta-actin (CAG) promoter and the green flouresecence protein (GFP)-encoding sequences in the AAV shuttle plasmid (pAAV-CAG-GFP, Addgene#37825) were initially replaced with a non-coding 32 bp DNA sequence containing unique restriction sites. To achieve this, 2 oligonucleotides containing *Not*I*, Sal*I*, Hin*dIII and *Eco*RV sites were synthesised, such that when annealed, they form a 32 bp duplex with *Nde*I and *Mfe*I overhangs. The 2 oligonucleotides (5′TATGCGGCCGCGATTCGTCGACAAGCTTC-3′ and 5′AATTGAAAGCTTGTCGCGATTCGCGGCCGC-3′), each at a concentration of 100 pM, were denatured at 95 °C for 5 min then allowed to cool to room temperature over 2 h. The resulting double-stranded DNA was ligated to pAAV-CAG-GFP, which in turn had previously been digested with *Nde*I and *Mfe*I. The resultant plasmid containing a multiple cloning site (MCS) was denoted pAAV-MCS.

To construct the AAV plasmids carrying greater-than-genome-length HBV sequences, pCDNA-1.28 × A1 or pCDNA-1.28 × D3 and pAAV-MCS were digested with *Not*I and *Sal*I. The HBV sequences (A1 or D3) were ligated to pAAV-MCS to generate pAAV-A1 and pAAV-D3, respectively. Correct insertion of the HBV fragment and intact inverted terminal repeats (ITR) sequences in the pAAV-A1 or pAAV-D3 constructs were confirmed using restriction analysis.

To assess the level of HBV gene expression from AAV plasmids carrying HBV genome sequences, the pAAV constructs (pAAV-A1 or pAAV-D3), pCH-9/3091 [27] or pAAV-empty were individually transfected into liver-derived Huh 7 cells using Polyethyleneimine “Max” (PEI-MAX) (Polyscience Inc., Warrington, PA, USA) according to the manufacturer’s instructions. Cells were then incubated at 37 °C with 5% CO_2_ in a humidified incubator for 48 h. The culture medium was harvested and the HBV surface antigen (HBsAg) concentration was measured using the Monolisa^®^ Ag HBs Plus immunoassay kit (Bio-Rad, Hercules, CA, USA) according to the manufacturer’s instructions.

### 2.2. Production of AAV Vectors

Recombinant AAVs of this study were pseudo-typed with serotype 2 (AAV2-HBV) or serotype 8 (AAV8-HBV) capsids for use in cultured cells and in vivo, respectively. To generate recombinant AAVs, triple transfection was carried out using PEI-MAX (Poly-science Inc, Warrington, PA, USA) according to the manufacturer’s instructions. Briefly, HEK 293 T cells were transfected with pAAV plasmids carrying HBV sequence of subgenotype A1 (pAAV-A1), subgenotype D3 (pAAV-D3,) a plasmid carrying an irrelevant sequence (pAAV-empty), a plasmid carrying primary microRNA sequences targeting HBV (pAAVMTTR 31/5,8,9, [28]) or a plasmid carrying primary microRNA sequences against hepatitis C virus (HCV) (pAAV-mTTR-BCDE, [29]). Each of the aforementioned plasmids were co-transfected with plasmids encoding AAV serotype 2 (AAV2) or AAV serotype 2 (AAV8) rep and cap genes [30] and the adenoviral helper protein (pXX-6, [30]). AAVs bearing the HBV replication-competent sequences were single stranded (ssAAVs), and those vectors encoding the artificial primary microRNAs (apri-miRs) were self-complementary (scAAVs).

At 48 and 72 h post-transfection, the spent medium was collected and replaced with the fresh medium. Then, 96 h after transfection, the spent medium was collected, and the cells were harvested and lysed with 3 cycles of freeze-thawing. Lysates were treated with benzonase (50 units/mL, Sigma, St. Louis, MO, USA) at 37 °C for 1 h, then clarified by centrifugation, and the supernatant was transferred to a sterile tube. Viruses from the spent medium were precipitated overnight using standard methods of precipitation with polyethylene glycol (PEG) 8000 (Sigma, St. Louis, MO, USA) and centrifugation. The PEG-precipitated viral pellet was resuspended with the supernatant from the cell lysate, and the virus then purified using iodixanol gradient ultracentrifugation and then stored at −80 °C. To quantify the yield of viral particles, AAV DNA was purified using a QIAamp DNA mini kit (Qiagen, Hilden, German, Germany). Purified DNA was analysed using quantitative PCR (q-PCR) with primer set HBVs-F and HBVs-R [31] and 2× FastStart essential DNA green master reaction mix (Roche, Basel, BS, Switzerland).

### 2.3. Assessment of HBV Gene Expression and Production of HBV Particles following Transduction of Cultured Cells with Recombinant AAV-HBV Vectors

To analyse HBV gene expression in vitro, Huh 7 cells were infected with AAV2 vectors carrying HBV subgenotype D3 (AAV2-D3), subgenotype A1 (AAV2-A1) sequences or the empty vector at an MOI of 10^4^. The supernatant was collected 48 h after infection and the HBsAg concentration was measured using the Monolisa^®^ Ag HBs Plus immunoassay kit (Bio-Rad, Hercules, CA, USA). Cells from the same experiment were used for in situ intracellular detection of HBV core antigen (HBcAg) using a mouse anti-HBcAg antibody (Abcam, Waltham, MA, USA) and a secondary Alexa Flour 488-labelled goat anti-mouse antibody (Thermo Fischer Scientific, Waltham, MA, USA). A standard immunofluorescence staining protocol was used, and counterstaining was used with 4′, 6-diamidino-2-phenylindole (DAPI) [32].

On separate duplicate plates, cells were transduced with AAVs at an MOI of 10^4^ and the supernatant was collected on days 1, 3 and 5 thereafter. The supernatant was subjected to DNA extraction using the QIAamp DNA mini kit (Qiagen, Hilden, German, Germany). The DNA was then used as the template for qPCR using the 2× FastStart essential DNA green master reaction mix (Roche, Basel, BS, Switzerland). The AcroMetrix^®^ HBV panel of HBV standards (Lifetech, Carlsbad, CA, USA) or 10-fold dilution of pAAV-A1 plasmid was used as standards for total or AAV genome copies quantification, respectively. HBVs-F and HBVs-R primers [31] were used to quantify total viral genome copies. The AAV-F (GCCATGCTCTAGGAAGATCGTACC) and AAV-R (CCGTAAATAGTCCACCCATTGACGT) primers were used to quantify AAV genome copies. The HBV DNA levels were determined by calculating the difference between the number of AAV genomes and total genomes.

### 2.4. Treatment of Mice with AAVs and Assessment of HBV Replication In Vivo

The Naval Medical Research Institute (NMRI) strain of female mice was used. Doses of 5 × 10^11^ or 9 × 10^11^ AAV8 vectors were administered to mice via the tail vein, and retro orbital puncture was used to collect blood. Serum was collected after centrifugation and stored at −80 °C before use. For tissue sampling, mice were euthanised using a rising CO_2_ chamber and liver tissues were collected and immersed in formalin before storage at 4 °C or frozen in liquid nitrogen before storage at −80 °C.

### 2.5. Evaluation of AAV-Induced Innate Immune Stimulation in Mice

To determine the safety of administering high and low doses of AAV8 vectors carrying the HBV subgenotype A1 (AAV8-A1) or subgenotype D3 (AAV8-D3) genomes, four mice per group were each injected with either 5 × 10^11^ or 9 × 10^11^ viral particle equivalents (VPEs). Mice in negative and positive control groups were injected with saline or 100 µg of polysinosic-polycytidylc acid (poly (I:C)), respectively. Serum collected at 6 h after injection was analysed using a cytometric bead array (CBA) mouse inflammation kit (BD Biosciences, Franklin Lakes, NJ, USA) to measure interleukin 6 (IL-6), interleukin 10 (IL-10), monocyte chemoattractant protein-1 (MCP-1), interferon gamma (IFN-γ), tumour necrosis factor alpha (TNF-α) and interleukin 12p70 (IL-12p70). The assay was carried out according to the manufacturer’s instructions and processed with a Fortessa FSR flow cytometer using a BD FACSDiva software, and data were analysed using FCAP Array software 3.0 (BD Biosciences, Franklin Lakes, NJ, USA).

To assess induction of an interferon (IFN) response, RNA was extracted from the livers of mice at 6 h after receiving the high (9 × 10^11^ VPEs/mouse) or low (5 × 10^11^ VPEs/mouse) doses of AAV. RNA purification made use of Tri Reagent^®^ (Sigma, St. Louis, MO, USA) according to manufacturer’s instructions. Primers specific for mRNAs encoding interferon beta (*IFN-β:* mIFN-*β*-F 5′CCACAGCAGCTTCTGACACTGAAAA3′ and IFN-*β*-R: 5′CGATCCCAGCCTTGTGTAACCAAATAACTTTTAATCGAA3′), Oligoadenylate synthase 1 (*OAS-1:* m*OAS-1*F 5′CGATTAAAAGTTATTTGGTTTCACAAGGCTGGGATCGAAAC3′ and m*OAS-1*R 5′CGATCCCAGCCTTGTGAAACCAAATAACTTTTAATCGAAGCA3′) and IFN-induced protein with tetratricopeptide repeats (*IFIT-1: IFIT-1F* 5′TAAAAGTTATTTGGCCGGACAAGGCTGGGATCGAAACC5′ and *IFIT-1R* 5′CGATCCCAGCCTTGTCCGGCCAAATAACTTTTAATAGA3′) were used to quantify respective mRNAs relative to murine *glyceraldehyde-3-phosphate dehydrogenase* (m*GAPDH:* m*GAPDH*-F 5′TAAAAGTTATTTGGCCTCACAAGGCTGGGATCGAAACC3′ and m*GAPDH*-R 5′CGATCCCAGCCTTGTGAGGCCAAATAACTTTTAATAGA3′). One microgram of RNA was subjected to reverse transcription quantitative PCR (RT-qPCR) using Luna^®^ Universal One-Step RT-qPCR Kit (New England Biolabs, Madison, WI, USA).

### 2.6. Measurement of HBV Surface Antigen, HBV Core Antigen and HBV e Antigen Expression in Mice

To assess HBV gene expression in mice following AAV administration, 8 mice per group were injected via the tail vein with AAV8 vectors at 5 × 10^11^ or 9 × 10^11^ VPEs/mouse. Control animals received saline. HBsAg expression was measured using serum collected at 2, 4, 8, 12 and 24 weeks after injection. The serum was diluted 1:4, and the assay was carried out using Monolisa^®^ Ag HBs Plus immunoassay kit (Bio-Rad, Hercules, CA, USA). HbeAg assay of serum diluted 1:12 was carried out by the National Health Laboratory Services (NHLS, Gauteng, Johannesburg, South Africa). The liver tissue samples of mice euthanised at 8 and 24 weeks post-injection were processed for HBcAg immunostaining (AMPATH laboratories, Rosebank, South Africa).

### 2.7. Detection of HBV Particle Equivalents, DNA Intermediates and RNA in Mice

To quantify the circulating HBV particle equivalents, DNA was extracted using the QIAamp DNA Blood Mini Kit (Qiagen, Hilden, Germany). HBV DNA concentrations in the murine serum samples were determined as described above. To detect intrahepatic HBV DNA replication intermediates, mice were euthanised at 24 weeks post-infection and total genomic DNA was extracted using a previously described method [33]. HBV DNA replication intermediates were detected by Southern blot, and the HBVs-1R oligonucleotide [31] was radioactively labelled and used to probe for relaxed circular (rcDNA), single stranded (ssDNA) and double stranded (dsDNA). To detect HBV transcripts, RNA was isolated from liver samples collected at 24 weeks post-infection, again using Tri Reagent^®^ according to the manufacturer’s instructions (Sigma, St. Louis, MO, USA). Thirty µg of RNA was analysed using Northern blot hybridisation with the radioactively labelled HBs-1R sequence as probe [32].

### 2.8. Assessment of Hepatotoxicity, Inflammatory Cell Infiltration and Fibrosis in Mice

Alanine aminotransferase (ALT) was measured as a marker of liver damage. The assay was carried out using an Advia 1800 Chemistry System (Siemens) at the accredited facilities of the South African National Health Laboratory Service (NHLS, Gauteng, Johannesburg, South Africa). Tested serum was collected at 2, 4 and 8 weeks after injection.

To assess the change in liver tissue histology, livers were harvested from mice sacrificed at 8 weeks and 24 weeks after injection. The liver tissues were processed for staining with haematoxylin and eosin (H&E) or Sirius red (AMPATH laboratories, Auckland Park, South Africa).

### 2.9. Assessment of Artificial Primary Micro-RNAs Efficacy against

#### HBV in AAV-HBV Model

To assess inhibition of HBV gene expression in vitro, Huh 7 cells were co-transduced with AAV2-D3 or AAV2-A1 and scAAV2-mTTR-BCDE or scAAV2 apri-miR 31/5,8,9, each at an MOI of 10^4^. Controls included cells treated only with AAV2-D3 or AAV2-A1. At 48 h after treatment, HBsAg levels in the supernatant were quantified using an Monalisa HBsAg ULTRA assay kit (Bio-Rad, CA, USA).

The safety in mice of co-delivering AAVs carrying HBV genomes and anti-HBV apri-miRs was assessed by co-administration of AAV8-A1 or AAV8-D3 at doses of 5 × 10^11^ or 9 × 10^11^, together with 1 × 10^11^ vector equivalents of scAAV8 apri-miR 31/5,8,9. Six hours post-co-injection, blood samples and liver tissues were collected. Serum was used to measure IL-6, IL-10, MCP-1, IFN-γ, TNF-α and IL-12p70 using a CBA mouse inflammation kit (BD Biosciences, Franklin Lakes, NJ, USA), Livers were used for RNA extraction then RT-qPCR to assay mRNA of *IFN-β*, *OAS-1* and *IFIT-1*, as described above.

To assess the use of AAV-HBV for assessing anti-HBV apri-miRs potency, mice were co-injected with AAV8-A1 or AAV8-D3 and scAAV8 apri-miR 31/5,8,9, as described above. The control group of mice received a vector carrying primary microRNA that targeted HCV (scAAV8 mTTR-BCDE [29]). Blood samples were collected at 2-, 8-, 12- and24- weeks post-co-injection. Serum concentration of HBsAg was quantified using Monalisa HBsAg ULTRA assay kit (Bio-Rad, Hercules, CA, USA). Circulating VPEs were measured by qPCR, as described above. Intrahepatic HBcAg was detected using immunohistochemistry, which was carried out at 24 weeks after vector administration and has been described above.

## 3. Results

### 3.1. AAV2-HBV Vectors Mediate HBV Gene Expression and Replication In Vitro

Recombinant DNA technology was used to incorporate greater-than-genome-length HBV sequences of subgenotype A1 and D3 into single-stranded AAV (ssAAVs), such that the HBV genome sequence was flanked by ITRs (Figure 1a). Compared to untransfected or empty control plasmid, Huh 7 cells transfected with AAV plasmids carrying HBV subgenotype A1 or D3 sequences resulted in significant HBsAg secretion at 48 and 72 h post-transfection (Appendix A). Transduction of Huh 7 cells with AAV2 vectors carrying HBV sequence of subgenotype A1 (AAV2-A1) or subgenotype D3 (AAV2-D3) vectors yielded significant HBsAg secretion as compared to uninfected and empty vector controls (Figure 1b). Similarly, HBcAg expression was detected in cells transduced with AAV2-A1 and AAV2-D3 but not in the controls (Figure 1c). When qPCR was used, HBV VPEs were detected in significant amounts after transduction with AAV2-A1 or AAV2-D3 (Figure 1d). These data demonstrate that the HBV genome-bearing AAVs successfully transduced liver-derived cells and led to HBV gene expression and viral particle production.

### 3.2. Administration of Recombinant AAV8-HBV Vectors to Mice Does Not Result in an Immunostimulatory Response

Deaths caused by high AAV vector doses have previously been reported in animals [34]. To assess the safety of both higher (9 × 10^11^ VPEs/mouse) or lower (5 × 10^11^ VPEs/mouse) doses used here, concentrations of serum pro-inflammatory cytokines and mRNA of intrahepatic IFN response genes were measured 6 h after the administration of saline or ssAAVs. CBA revealed that serum concentrations of IL-12p70, IL-6, TNF-α, IL-10, MCP-1 and INF-γ were similar in mice receiving AAV8 vectors carrying the HBV sequence of subgenotype A1 (AAV2-A1) or subgenotype D3 (AAV2-D3) and saline (Figure 2a–f). Comparable findings were observed following RT-qPCR carried out on the mRNA of IFN response genes (*OAS-1*, *IFN-β* and *IFIT1*) (Figure 2g–i). As expected, the levels of all the cytokines in mice injected with poly (I:C) were significantly higher compared to mice injected with the empty vector or saline (Figure 2a–i). Taken together, these results confirm an established property of AAVs, which is that when administered at lower doses, they have low immunogenicity and are unlikely to induce adverse innate immune responses.

### 3.3. Transduction of Mice with AAV8-HBV Vectors Results in HBV Gene Expression and Replication

To assess AAV8 vector-mediated HBV gene expression, serum levels of HBsAg were measured at 2, 4, 8, 12 and 24 weeks after mice received AAV8-A1 or AAV8-D3. Sustained expression of HBsAg was detected in mice receiving either AAV8-A1 or AAV8-D3. A dose-dependent response was observed in mice treated with AAV8-A1, but AAV8-D3-infected mice showed similar HBsAg levels at both doses (Figure 3a). The levels of HBeAg were also measured, but low volumes of available serum samples limited accurate detection (data not shown). To determine whether the ssAAV vectors are capable of mediating the production of HBV particles, circulating VPEs were quantified by qPCR using DNA extracted from serum samples collected at various time points after AAV8 treatment. High concentrations of VPEs were observed in the serum of mice injected with AAV8-A1 and AAV8-D3 but were undetectable in control mice receiving the empty vector or saline (Figure 3b). Immunohistochemical analysis to detect intrahepatic HBcAg confirmed that AAVs efficiently mediated HBV gene expression at 8 and 24 weeks after the animals received AAVs. Surprisingly, the HBcAg was mainly located in the hepatocyte nuclei of mice receiving AAV8-A1. Conversely, HBcAg was predominantly detected in the cytoplasm of mice receiving AAV8-D3 (Figure 3c).

To detect different forms of intrahepatic HBV DNA (rcDNA, dsDNA, ssDNA and cccDNA) and viral transcripts (3.5 kb, 2.4 kb and 2.1kb RNAs), Southern and Northern blotting were performed, respectively. Analyses were carried out on DNA or RNA isolated from hepatic tissue of mice at 24 weeks after the animals had each received 9 × 10^11^ VPEs of AAV8-A1 or AAV8-D3. rcDNA, dsDNA (Figure 4a) and HBV RNA transcripts of 3.5 kb, 2.4 kb and 2.1kb in size were detectable (Figure 4b). These results demonstrate that AAV8 mediates production of key intermediates essential for HBV replication.

### 3.4. Injection of Mice with AAV8 Vectors Bearing HBV Sequences Does Not Cause Hepatotoxicity or Fibrosis

The serum activity of ALT was measured to determine whether the HBV gene expression and replication in AAV8-HBV-transduced mice led to liver toxicity. At two weeks post-administration of AAV8-A1 and AAV8-D3, ALT levels were higher compared to mice injected with saline or AAV-empty and equal between both subgenotypes. This suggests that HBV gene expression may result in minor toxicity. However, ALT activity gradually declined and by week 8, the levels were similar to those observed in mice injected with saline or AAV-empty (Figure 5a). ALT levels in all mice remained below the accepted threshold of normality (<100 IU/L), suggesting there was no significant onset of liver injury as a result of AAV administration.

The liver sections prepared from mice at 24 weeks post-injection were used for H&E or Sirius red staining. H&E revealed minimal inflammation in mice treated with AAV8-D3, and not in those transduced with either AAV8-A1, AAV8-empty or saline. Sirius red staining did not demonstrate fibrotic effects as a result of AAV8 administration (Figure 5b). Taken together, these data suggest that administration of AAV8-HBV at optimal doses does not cause significant toxicity.

### 3.5. Successful Use of AAV8-HBV Model to Asess the Efficacy of Anti-HBV Pri-miR

Evaluating the AAV8-A1 model for the use in assessing the efficacy of anti-HBV gene silencers was performed in both cell culture and in vivo. The level of HBsAg was measured in the supernatants of Huh 7 cells co-transduced with AAV2-A1 or AAV2-D3 and AAV2 expressing apri-miRs against HBV or HCV, both at an MOI of 10^4^. Compared to the controls, HBV gene silencing of more than 90% was observed at 48 h after treatment with the vector carrying anti-HBV sequences (Appendix A). It has previously been reported that administering AAV8 expressing artificial gene silencers to mice caused severe toxicity that led to fatalities [30]. The toxicity was later linked to the oversaturation of Agonaute-2, an essential mediator of RNAi, following the overexpression of the exogenous RNAi activators [35]. To evaluate whether the co-administration of AAV serotype 8 vectors carrying anti-HBV primary micro RNA (scAAV8 apri-miR 31/5,8,9) and AAV8 carrying HBV genome (AAV8-HBV) vectors induced adverse immune events and liver toxicity, serum cytokine concentration and interferon response genes expression were quantified. Measurements were made 6 h after administration of AAV8s, poly (I:C) or saline. Compared to mice injected with saline, there were no significant differences in the concentrations of the measured serum cytokines (IL12p70, IL-6, TNF, IL-10, MCP-1 and INF-γ) or mRNA from inflammatory response genes (*OAS-1*, *IFN-β* and *IFIT-1*). As expected, the immunogenic poly (I:C) caused robust immune stimulation of cytokines and IFN response genes (Appendix A). ALT levels determined over a period of 8 weeks in mice co-administered with AAV8-HBV and anti-HBV vector were within normal limits, and similar to activities measured in saline-injected mice (Appendix A).

To evaluate the silencing of HBV gene expression in vivo, mice were given a combination of AAV8-A1 or AAV8-D3 and self-complementary AAV8 vectors carrying anti-HBV primary micro-RNA (scAAV8 apri-miR 31/5,8,9) or self-complementary AAV8 vectors carrying anti-HCV (scAAV8-mTTR-BCDE) vectors. Compared to mice receiving the control scAAV8-mTTR-BCDE, scAAV8 apri-miR 31/5,8,9 effected a significant knockdown of HBsAg expression of about 97% (Figure 6a). This inhibitory effect was sustained over a period of 24 weeks. Similar results were observed when HBV VPEs were quantified from the serum of the co-injected mice (Figure 6b). The efficacy of the anti-HBV RNAi activator in AAV8-HBV-transduced mice was further demonstrated by immunohistochemical staining and quantification hepatocytes that were positive for HBcAg (Figure 6c). Together, these data show that the AAVs are valuable for modelling the replication of different HBV subgenotypes in vivo. Moreover, use of ssAAVs carrying HBV sequences may be used conveniently to assess the efficacy of new candidate anti-HBV drugs.

## 4. Discussion

Chronic HBV infection continues to be a global health problem with serious consequences, including cirrhosis and liver cancer. Annual HBV-related deaths are estimated to be approximately 820,000. Sub-Saharan Africa, including South Africa, bears a significant burden of the disease. South Africa, in particular, is endemic for subgenotype A1, which is associated with more aggressive liver cancer [36]. Advancing effective modes of treating chronic HBV infection and achieving functional cure are a priority of research in the field [3]. Access to physiologically relevant and convenient small animal models is vital for assessing novel therapeutic approaches to clinical application. The death of animal models that simulate all steps of HBV replication has hampered progress with HBV therapy. Previous studies recapitulated chronic HBV genotype D infection after transduction of murine hepatocytes in vivo with engineered ssAAVs [23,24]. Subgenotype A1 has unique molecular characteristics differentiating it from genotype D both in vivo and in vitro [7,37,38,39]. Developing models that analyse the replication of other genotypes, particularly subgenotype A1, is therefore important to assess responses to treatment with new candidate therapies.

In this study, we have shown that the incorporation of greater-than-genome-length HBV genotype A1 sequences into ssAAVs could be used to achieve sustained HBV replication in vivo. AAVs have several well-established advantages for use in gene therapy and other applications requiring the transduction of cells. For example, methods for their propagation are now well established; furthermore, they are safe and cause minimal innate immunostimulation. These properties are suited to the modelling of HBV replication and flexibility enables adaptation in investigating the replication properties of different HBV genotypes. Adenoviral vectors can deliver HBV sequences and induce replication in murine hepatocytes in vivo [20], but strong induction of an immune response in this model is a drawback. HBV itself is a ‘stealth’ virus that has minimal effects on innate immunity [40,41]. Ideally, vectors carrying HBV sequences should therefore also have limited immunogenicity to model replication accurately. AAVs do cause some induction of innate and adaptive immune responses through interaction with pattern recognition receptors (PRRs) such as toll-like receptor (TLR)-2 and TLR-9 [42]. In mice, AAV genome configuration may also influence the innate response; self-complementary AAV genomes increase transgene immunity, whereas the response following ssAAV administration is attenuated [43]. Our data demonstrate minimal induction of an immune response with the AAVs encoding HBV subgenotype A1 sequences and apri-miRs. The use of vectors at doses of either 5 × 10^11^ or 9 × 10^11^ VPEs/mouse safely deliver HBV sequence with no evidence of induction of inflammatory response (Figure 2a–i).

AAV doses of 5 × 10^10^ or 2 × 10^11^ VPEs/mouse were previously reported to deliver greater-than-genome-length HBV genotype D to C57BL/6 mice [24]. Significant HBV gene expression and replication was demonstrated for 6 months to a year following a single systematic administration of the AAVs. In the study reported here, we found that the injection of mice with 1 × 10^9^ or 1 × 10^10^ VPEs/mouse resulted in no detectable HBV gene expression (data not shown), whereas the dose of 1 × 10^11^ VPEs/mouse resulted in very low HBsAg expression (Appendix A). Reasons for this discrepancy are not yet clear. A possible explanation is the differences in strains of mice or possibly the different subgenotypes of HBV that were used in the two studies. Different mouse strain-dependent HBV gene expression has previously been reported [22,44]. Here we observed high-level HBV gene expression and replication when mice were injected with 5 × 10^11^ or 9 × 10^11^ AAV vectors bearing HBV sequences of subgenotype A1 or D3. Elevated HBsAg, HBcAg and HBV VPEs were sustained over a period of 6 months. Compared to subgenotype D3, subgenotype A1 HBV sequences resulted in approximately 10-fold lower HBV gene expression after administration of similar vector doses (Appendix A). This differential gene expression between the two genotypes was also evident after injecting a higher dose of ssAAVs (5 × 10^11^ VPEs/mouse) (Figure 3a,c) and supports previous studies showing differences between these subgenotypes in patients and following the transfection of hepatoma cells with replication competent clones [37,38,39]. Although HBV particle production is demonstrated, HBV infection of hepatocytes does not occur and cccDNA formation in murine hepatocytes is limited. However, resulting prolonged HBV gene expression mimics chronic human infection, which is a useful feature of the model.

Immunohistochemical staining revealed that HBcAg was mainly located in the nuclei of hepatocytes in mice treated with AAV8-A1, whereas in mice injected with AAV8-D3, HBcAg presence was more pronounced in the cytoplasm (Figure 3c). HBcAg bears a nuclear localization signal [45] and is expected to locate to the nucleus. Both viral and host factors can play a role in the subcellular localization of HBcAg [46]. The differential gene expression between subgenotypes A1 and D3, seen in the present study (Appendix A and Figure 3a) and other studies [26,39], may account for the differences in subcellular localization of HBcAg seen in the mouse. Other viral factors that have been implicated in influencing HBcAg subcellular localization include the sequence variation in the basic core promoter (BCP) [47] and the core C-terminus arginine rich region [48]. Apart from the direct repeat insert at positions 153–154 of core, the characteristic of genotype A, the C-terminus is entirely conserved between subgenotypes A1 and D3. However, the BCP of subgenotype A1 has unique molecular characteristics differentiating it from other subgenotypes, including D3 [9,49]. Dominant cytoplasmic HBcAg localization has previously been associated high HBV DNA replication, whereas nuclear localization was correlated with reduced viral DNA synthesis [50] and reinforces our observation that markers of HBV replication were higher in mice receiving the subgenotype D3 than in those animals treated with ssAAV A1 (Appendix A and Figure 3a). The subcellular localization of HBcAg is cell cycle-regulated [51], with nuclear expression of HBcAg predominantin quiescent and resting cells [52] with normal or minimal histological activity [53] and in the liver of patients with non-aggressive disease [54].

Although HBV particles from the AAV-HBV model are not expected to be different to new HBV particles from a natural HBV infection, it remains to be confirmed whether HBV particles produced in this study can infect human cells and enable cccDNA production. Difficulties with murine models of HBV replication are that murine hepatocytes are not naturally susceptible to HBV infection, and the cccDNA is normally not produced in mouse liver cells. Surprisingly, a recent study detected cccDNA 4 weeks following the infection of mice with AAV-HBV carrying HBV genotype D genome [55]. In this study, only rcDNA and dsDNA were detected (Figure 4a). Interestingly, during the construction of this manuscript, a new study supported the notion of cccDNA production in AAV-HBV model. They demonstrated that the delivery of either HBV genotype A, B, C or D genome produces cccDNA in mice hepatocytes over a period of 8 weeks. The cccDNA produced was functional and indistinguishable in terms of sequence from cccDNA produced after HBV infection in culture. However, the cccDNA levels peaked at 1 week and gradually decreased over time, with only 28% remaining at 8 weeks post-infection. This could explain why we did not detect cccDNA at 24 weeks post-infection and necessitates analyses at an earlier time point. The study also showed that HBV gene expression peaked at 2 weeks and was sustained over 8 weeks period, suggesting that cccDNA contribution to HBV gene expression in this model is minor. Whereas the mechanism of its formation needs to be investigated further, it is suggested to be a product of intramolecular recombination between HBV genome ends and is HBV replication-independent [56].

In our study, mild features of inflammation were observed in mice injected with AAV carrying subgenotype D3 sequences but not subgenotype A1 genome-treated mice. Clear hepatotoxicity and liver fibrosis were not observed in mice receiving either of the subgenotype sequences (Figure 5). Reports on liver inflammation, injury and fibrosis following the infection of mice with AAVs carrying genotype D genome has been contradictory [22,23,24,57]. Some studies reported inflammation, HCC, fibrosis and chronic liver damage, but other studies did not replicate these findings. Different HBV strains, genetic backgrounds of mice and/or AAV doses used may contribute to variable observations. However, the absence of significant inflammation reported here and in previous studies suggests HBV avoidance of immunity and possible T cell exhaustion [22,58,59]. The mechanisms underlying fibrosis and chronic liver damage that have been previously described in AAV-HBV murine models remain to be established [60,61,62,63]. Differences in viral antigenaemia and HBV DNA integration as well as host immune responses may contribute to the variability.

Newly developed gene therapy-based approaches, such as those based on RNAi activation and gene editing, have shown potential for eradicating HBV infection. Susceptibility of HBV to inactivation based on RNAi has been extensively demonstrated [28,64,65], and the gene silencing approach is now being evaluated in phase 2 clinical trials (https://ir.arrowheadpharma.com/news-releases/news-release-details/arrowhead-and-collaborator-janssen-present-phase-2-clinical-0, date accessed: 9 September 2021). The analysis of the model described here verified the sensitivity of HBV genotype A1 to RNAi-based silencing. Because sequence variation among HBV genotypes and subgenotypes could influence silencing efficacy [66], it is important that anti-HBV-silencing therapeutic regimens be tested against a variety of HBV genotypes, such as the important subgenotype A1. This study is the first successful demonstration of sustained HBV A1 and D3 replication in a murine model following administration of engineered ssAAVs, with differences between the two subgenotypes. Moreover, sustained HBV viral replication and lack of significant immune stimulation through AAV-mediated delivery of HBV genomes is advantageous as it mimics chronic HBV infection closely. This is important to assess the potency of new antivirals and supports the utility of RNAi activators in disabling HBV replication. Successfully targeting the *X* ORF of HBV is important, because HBx is required for transcriptional function of cccDNA [67]. An added advantage is that the AAV HBV model may be used to test vaccine efficacy [23]. Overall, the AAV8-A1 mouse model is a significant addition to the toolbox required for advancing effective therapy to counter a serious global health problem.

## Figures and Tables

**Figure 1 viruses-13-02247-f001:**
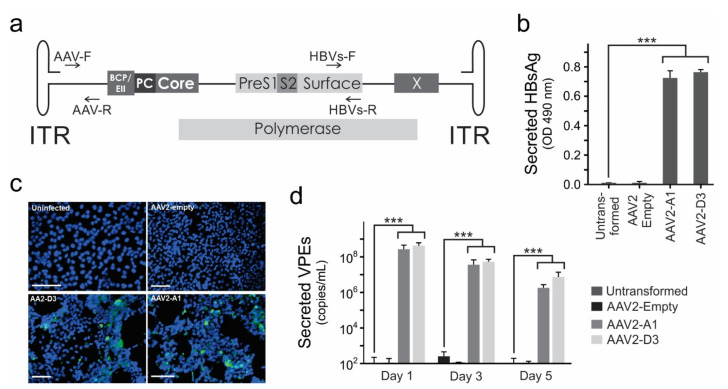
Hepatitis B (HBV) gene expression in liver-derived cells transduced with HBV genome-bearing single stranded adeno associated viral vectors (ssAAVs). (**a**), Schematic illustration of ssAAV genome used to deliver HBV greater-than-genome-length of subgenotype A1 or D3. The PCR primers for quantification of adeno associated viral vector (AAV) viral particle equivalents (VPEs) (AAV-F and AAV-R) and total VPEs (HBVs-F and HBVs-R) are indicated. (**b**), HBV surface antigen (HBsAg) expression in Huh 7 cells at 48 h post-transduction with AAVs at an molecule of infection (MOI) of 1 × 10^4^. (**c**), Immunofluorescence staining for HBV core antigen (HBcAg) in Huh 7 cells transduced with ssAAVs. Nuclei were counterstained with 4′, 6-diamidino-2-phenylindole (DAPI) and cells imaged at 20× magnification. Scale bars of 200 µm are shown. The HBcAg detection is shown in green. (**d**), HBV VPEs secreted from Huh 7 cells after transduction with AAVs at an MOI of 1 × 10^4^. Values represent the means and standard errors of mean (SEM) derived from 4 replicates. Statistically significant differences between samples were calculated using the Student’s two-tailed paired *t*-test. A statistically significant *p* value of ≤0.001 is indicated (***).

**Figure 2 viruses-13-02247-f002:**
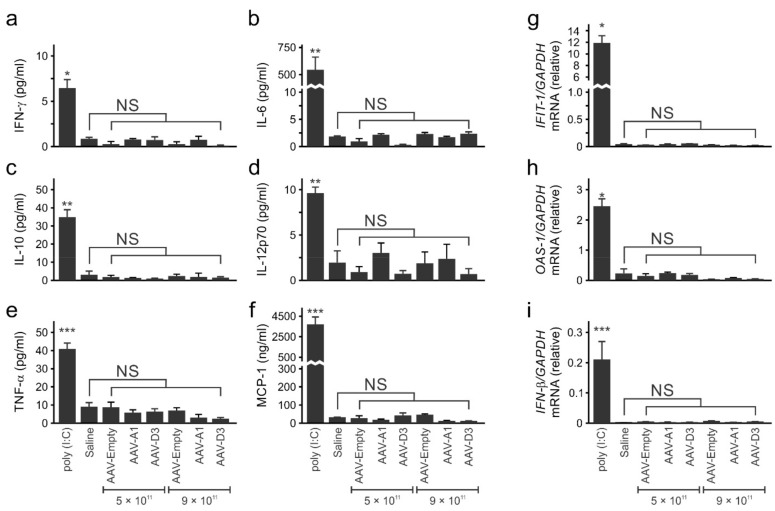
Quantification of cytokines and immune response gene expression in mice after injection with AAVs. Protein levels of (**a**), interferon gamma (IFN- γ), (**b**), interleukin 6 (IL-6), (**c**), interleukin 10 (IL-10), (**d**), interleukin 12p70 (IL-12p70), (**e**), tumour necrosis factor alpha (TNF-α) and (**f**), chemoattractant protein 1 (MCP-1) 6 h post-injection of AAVs, Poly (I: C) or saline. (**g**–**i**), *Oligoadenylate synthase 1* (*OAS-1)*, *interferon beta* (*IFN-β)* and *IFN-induced protein with tetratricopeptide repeats 1* (*IFIT-1)* gene expression relative to murine *glyceraldehyde-3-phosphate dehydrogenase* (*GAPDH)* gene expression in mouse livers collected 6 h post-injection with AAVs, saline or polysinosic-polycytidylc acid (poly (I:C)). Values represent the means and SEM calculated after injection of 4 mice per group. Statistically significant differences between samples were calculated using the Student’s two-tailed paired *t*-test. *p* values of ≤0.05 (*), ≤0.01 (**) and ≤0.001 (***) were considered statistically significant, NS: not-significant.

**Figure 3 viruses-13-02247-f003:**
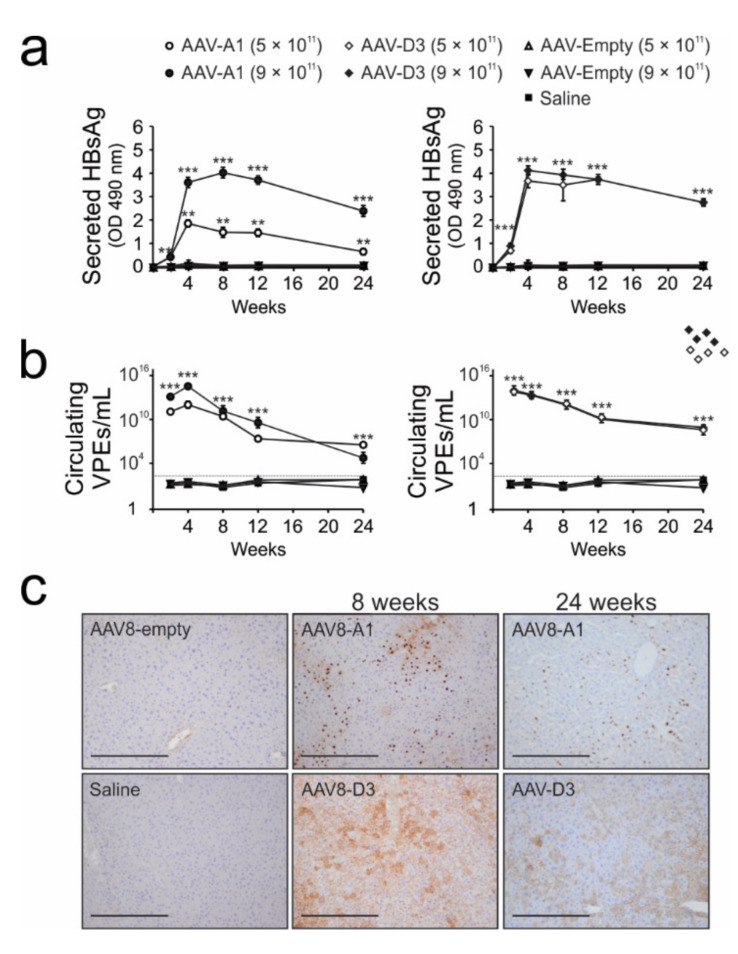
Evaluation of long-term HBV gene expression and replication in vivo. (**a**), HBsAg levels in serum collected from mice treated with saline or AAVs. Means and SEMs were calculated from data obtained from 8 mice per group. (**b**), serum concentrations of HBV VPEs after injection of mice with saline or AAVs. Values represent the means and SEMs calculated from data obtained from 4 mice per group. Statistically significant differences between samples were calculated using the Student’s two-tailed paired *t*-test. *p* values of ≤0.01 (**) and ≤0.001 (***) were considered statistically significant. (**c**), representative images showing HBcAg-immuno-positive hepatocytes in mice at 8 and 24 weeks after treatment with saline or AAVs at a dose of 9 × 10^11^ VPEs per mouse. Staining was performed on tissue sections from 3 mice per group. Brown stain shows HBcAg-positive cells at 10× magnification. Low-power fields with scale bars of 200 µm are shown.

**Figure 4 viruses-13-02247-f004:**
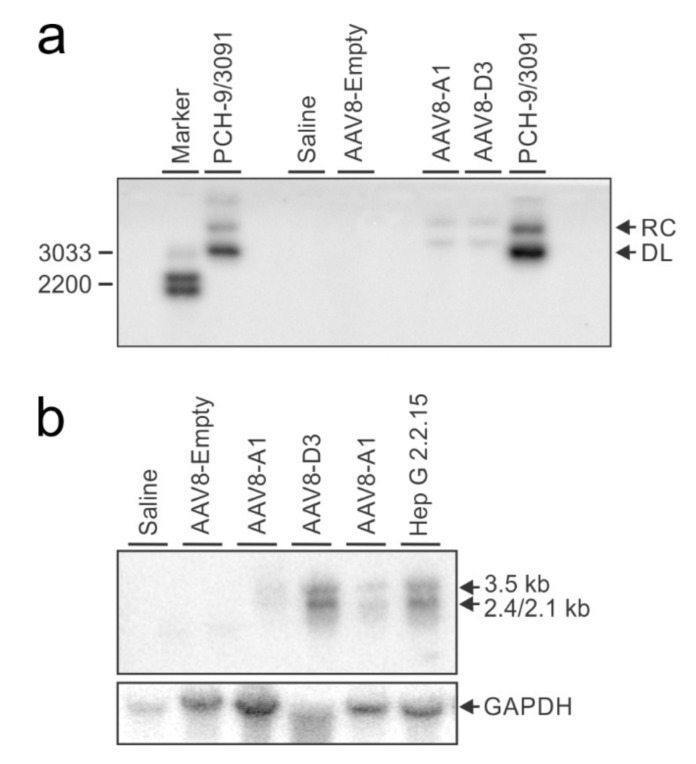
Detection of HBV replication intermediates. (**a**), HBV DNA intermediates from livers of mice extracted 24 weeks post-injection with saline or 9 × 10^11^ AAV VPEs/mouse. The pCH-9/3091 plasmid carrying HBV genotype D sequence was included as a control to verify probe hybridisation. Arrows indicate relaxed circular (RC) and double-stranded linear (DL) DNA. (**b**), RNA intermediates from the livers of mice extracted 24 weeks post-injection with saline or 9 × 10^11^ AAV VPEs/mouse. RNA extracted from Hep G 2.2.15 cells was included as the positive control. Arrows indicate pre-genomic RNA (3.5 kb) and sub-genomic RNAs (2.4 kb and 2.1 kb). The blot was re-probed with a *GAPDH*-specific probe to verify equal loading of the lanes.

**Figure 5 viruses-13-02247-f005:**
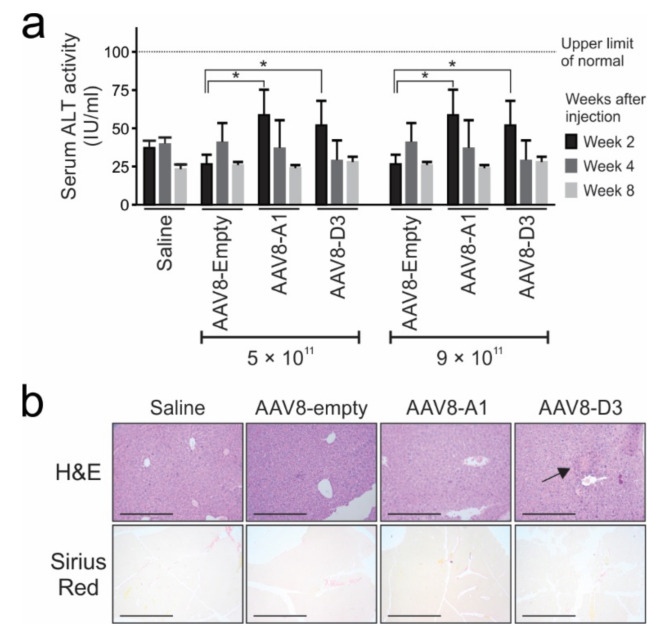
Assessment of inflammation and toxicity in liver tissues from AAV8 HBV-infected mice. (**a**), Alanine transaminase (ALT) activity in serum collected from mice treated with saline or AAVs. Means and SEM were calculated from data obtained from 4 mice per group. Statistically significant differences between samples were calculated using Student’s two-tailed paired *t*-test. *p* values of ≤0.05 (*) were considered statistically significant. The black dotted line indicates threshold of accepted upper limit of the normal range (<100 IU/L). (**b**), Haematoxylin and eosin (H&E) and Sirius red staining of liver sections collected at 24 weeks post-injection with AAVs or saline. Low-power fields with scale bars of 200 µm are shown, and tissues were visualised at 40× magnification. The black arrow indicates minimal inflammation, with blue dots representing the nuclei of infiltrating mononuclear lymphocytes.

**Figure 6 viruses-13-02247-f006:**
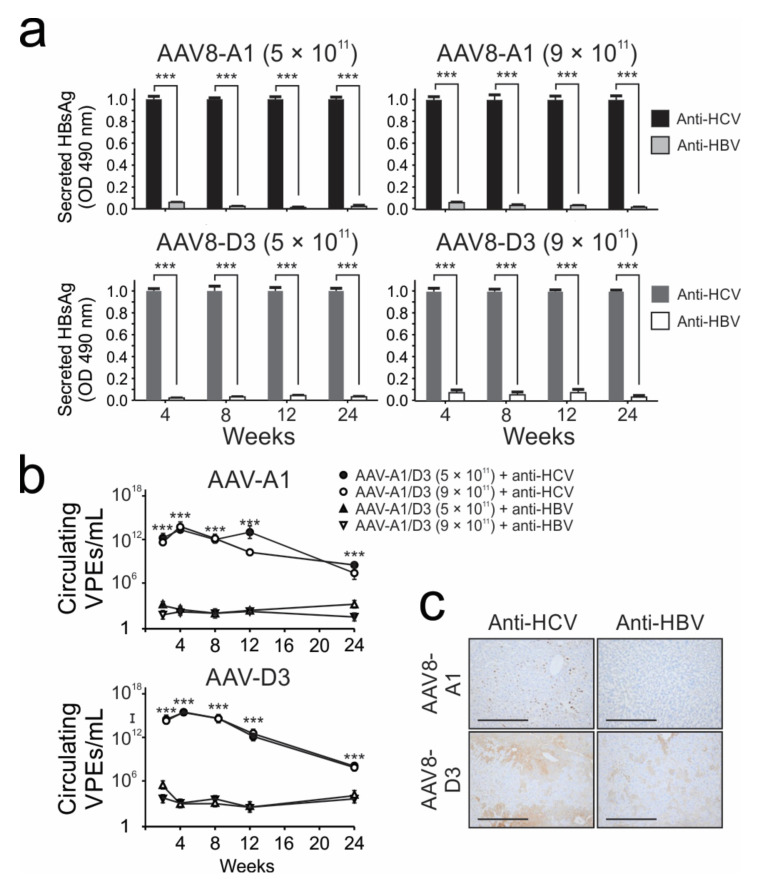
Inhibition of HBV gene expression and replication in mice treated with AAVs carrying HBV genomes and HBV-targeting artificial micro RNAs. (**a**), serum HBsAg and (**b**), HBV VPEs post-injection of mice with saline or co-transduction with AAV8-HBV and artificial primary microRNA (apri-miR)-expressing vectors. Values represent the means and SEMs calculated from data obtained from 4 mice per group. Statistically significant differences between the samples were determined using Student’s two-tailed paired *t*-test. A statistically significant *p* value of ≤0.001 was indicated (***). (**c**), Representative images of cells positive for HBcAg detected in the livers of animals assessed 24 weeks post-co-injection with AAV8-HBV and apri-miR expressing vectors. Low-power fields with scale bars of 200 µm are shown, and tissues were visualised at 20× magnification. Staining was performed on tissue sections from 3 mice per group.

## Data Availability

The data presented in this study are available in the main article or Appendix A.

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
