# Peer review of "In Vivo Modelling of Hepatitis B Virus Subgenotype A1 Replication Using Adeno-Associated Viral Vectors"

_viruses, 2021, doi:10.3390/v13112247_

Round 1
Reviewer 1 Report
This study focused on modeling HBV subgenotype A1 and D3 replication in culture cells and murine model as well as investigating anti-HBV therapy in AAV8-A1 murine model. The authors constructed recombinant AAV-HBV either transduced into HuH-7 cells or administered to mice. They revealed that it is feasible to express HBV genes and produce viral particles in vitro and in vivo; in addition, with very low immune response and no inflammation or toxicity appeared in mice. Notably, they applied anti-HBV apri-miRNA in AAV-HBV system and found that HBsAg, HBV replication, and HBcAg expression were diminished.
Overall, the data is promising, but the novelty of this study is relatively weak because the AAV model has been used in previous studies of HBV. Noted that the authors exaggerated that they successfully used the AAV model to screen anti-HBV therapeutics, which does not show in the presented study (using miRNA is not a screening). The authors should at least confirm the feasibility of such AAV-HBV model with conventional antiviral drugs. In addition, errors of labels in figures and figure legends should be revised before it can be accepted for publication.
Major issues:
- The study claims that AAV model can be used to “screen” anti-HBV therapeutics and anti-HBV drugs. However, no data support such idea. The result section: “Successful use of AAV8-HBV model to screen candidate anti-HBV therapeutics,” only assessed the inflammatory response and the viral product inhibition upon co-treatment with AAV and miRNA, which does not imply a therapeutic or drug screening. As a drug screening model, the author should test antiviral efficacy of conventional anti-HBV NUCs in the current model as a proof of concept.
- HBeAg levels should be provided in vitro and in vivo.
- The presented study attacks the weakness of current HBV models, including chimpanzees, NTCP-expressing hepatocytes, transgenic mice, and suggests the AAV model as an alternation. The authors should discuss the advantage and disadvantage of the current model in comparison with other existing ones.
Minor issues:
- The result section should be self-explained. The authors should clarify the definitions, methods, and agents instead of using abbreviations such as greater-than-length HBV sequence, pAAV-TTR-BCDE, spri-miR 31/5, scAAV8, etc.
- The context in discussion from lines 479 to 496 is hard to follow. The authors should refine the discussion section.
- Figure 3b right panel illustrated 5x1011 and 9x1011 VPEs of AAV8-D3 but the icons were circle. Those should be rhombus.
- The images should have scale bar: Figure 1c, 3c, 5b, 6c, and the fold magnifications should be mention in Figure legend 5b and 6c.
- The format of figure legends should be consistent.
- The figure caption in figure 4 is boldface.
- Page 10 line 370, 374 and page 11 line 412, 416: a, b, c, should be boldface.
- Page 7 line 286 and page 8 line 337: the Student’s two tailed paired “t-” test, t- should be added.
- Page 10 line 373: The red dotted line should be the “black” dotted line.
- Page 7 line 286 and page 11 line 416: P value “was” indicated, use the past tense.
- In supplementary data figure legend:
- S1: Data “were” expressed…, not “is” expressed.
- S2: AAV2-HBV MOI should be 1x104 instead of 1x105? Page 10 line 381 mentioned MOI of 104.
- S3: legend a-f are not corresponded to figure a-f.
Author Response
Dear Reviewer
Thank you very much for a thorough review of our manuscript and all comments are acknowledged. See attached the response to the comments.
Regards

Reviewer 2 Report
Despite the availability of a vaccine to prevent transmission of hepatitis B virus (HBV), complications resulting from chronic HBV infection present a vital global health problem. About 700,000 people die each year from HBV-related cirrhosis and hepatocellular carcinoma (HCC). Currently, licensed therapies rarely eliminate HBV, and the development of effective drugs for the treatment of HBV infection is a priority to limit the deaths of 240 million chronic carriers worldwide.
Approved HBV infection treatment regimens are mostly limited to interferon and nucleoside analogs, but these drugs can only effectively suppress viral replication without eliminating it.
The significant features of HBV, including a narrow host range and a strong tropism of the virus towards hepatocytes, have led to serious problems in developing a suitable model system for the HBV replication cycle. In addition to humans, chimpanzees (Pan troglodytes) and macaques (Macaca fascicularis), as well as Malay tupaya (Tupaia belangeri) are susceptible to HBV infection. All these animal models are either very expensive or cannot be used for ethical reasons. Therefore, where possible, it is preferable to use an in vitro model. The lack of a reliable and reproducible in vitro cell culture system capable of supporting all stages of the HBV replication cycle, including infection and cccDNA formation, also made it difficult to study the early stages of viral-cell interaction and development of anti-HBV drugs. It is also important that the systems used to monitor the antiviral efficacy of different agents are able to test the response of different subgenotypes of the virus to ensure that the antiviral method is pangenotypic and can be used in all regions of the world, especially where HBV is endemic, such as Africa.
In this peer-reviewed paper recombinant adeno-associated viruses (AAV) were used to model the replication of HBV subgenotypes A1 and D3 in vitro and in vivo. Recombinant vectors of the second and eighth serotypes, AAV2 and AAV8, bearing greater-than-genome-length sequences of HBV DNA were obtained from subgenotypes A1 and D3. Recombinant AAVs with serotype 2 (AAV2-HBV) or serotype 8 (AAV8-HBV) have been used in vitro and in vivo, respectively.
The resulting model supported long-term HBV replication mimicking natural chronic infection.
Compared to an untransfected or empty control plasmid, Huh7 cells transfected with AAV plasmids carrying HBV sequences of subgenotype A1 or D3 secreted HBsAg 48 and 72 hours after transfection. Likewise, HBcAg expression was found in cells transduced with AAV2-A1 and AAV2-D3, but not in controls, and HBV viral particle equivalents (VPE) were detected in significant amounts after transduction with AAV2-A1 or AAV2-D3. These data demonstrate that AAVs carrying the HBV genome successfully transduce liver cells and induce HBV gene expression and viral particle production. In the study, only HBV rcDNA and dsDNA were detected. The authors explain this by the fact that according to the literature, cccDNA levels peak after 1 week and gradually decrease over time, so no cccDNA was detected 24 weeks after infection. In this regard, it would be interesting to evaluate the formation of cccDNA at earlier stages.
HBsAg expression was also found in mice treated with AAV8-A1 or AAV8-D3.
High concentrations of VPE were observed in the blood sera of mice injected with AAV8-A1 and AAV8-D3.
Cytometric studies showed that serum concentrations of IL-12p70, IL-6, TNF-α, IL-10, MCP-1 and INF-γ were the same in mice treated with AAV8-A1, AAV8-D3 and saline, that is AAVs had low immunogenicity and did not appear to induce adverse activation of the innate immune system.
Interestingly, HBcAg was mainly localized in the hepatocyte nuclei of mice treated with AAV8-A1. In contrast, HBcAg was predominantly found in the cytoplasm of mice treated with AAV8-D3.
These results demonstrate that AAV8 mediates the production of key intermediates required for HBV replication.
In mice injected with AAV8, ALT levels were higher compared to mice injected with saline, and were the same for both subgenotypes. This suggests that HBV gene expression may result in minor toxicity. However, ALT activity gradually decreased, and by 8 weeks its levels were like those observed in mice that were injected with saline or empty AAV. Liver sections obtained from mice 24 weeks after injection showed minimal inflammation in mice treated with AAV8-D3, but not in mice transduced with AAV8-A1, AAV8-empty, or saline, and showed no fibrotic changes following AAV8 administration.
Thus, the resulting model mimics chronic human HBV infection. This study is the first successful demonstration of robust HBV A1 replication in a mouse model following administration of engineered AAVs. Moreover, the stable replication of the HBV virus and the absence of significant immune response stimulation makes the model very promising for testing some types of new antiviral drugs that affect the transcription and translation of viral proteins in the cytoplasm and nucleus, particularly, artificial primary miRNAs.
The data obtained in this work are of interest for infectious disease specialists and hepatologists. The research was carried out using adequate methods and the manuscript may be published.
Author Response
Dear Reviewer
Thank you for a thorough review of our manuscript and all comments are acknowledged. Attached is the response to the comments.
Regards
Betty

Reviewer 3 Report
Chronic HBV infection is a global health problem. In 2019 about 820,000 people die from HBV-caused cirrhosis and liver cancer. Sub-Saharan Africa bears a significant burden of the disease highly related to infection with HBV subgenotype A. However, there is lack of mouse HBV model, as the viruses do not naturally infect murine hepatocytes. The researchers have developed AAV vectors carrying HBV A1 genome, facilitating the studies of HBV-A infection mechanism and treatments with the mouse model. Their study, though previously having a report of AAV-HBV-D vector model, is still important. The experiments are also well performed.
The major comment
- As the AAV-HBV vectors can induce HBV infection and amplification in mice, the HBV from mice may somehow infect human after replication and possible accumulation of mutations. Considering current covid pandemic, do the authors have any safety concern and what can be done? These should be added in the Discussion.
Minor comments
- Line 100-102: "To allow insertion 100 of the HBV sequences into an AAV shuttle plasmid (pAAV-CAG-GFP, Addgene#37825), the CAG promoter and the GFP-encoding sequences were initially replaced with a non-coding 32 bp DNA sequence containing unique restriction sites."
It may cause confuse what sequences are deleted, from HBV or in AAV. It seems in AVV plasmid. It may be better to add "in the AAV plasmid" after "the CAG promoter and the GFP-encoding sequences".
- Line 212: "Measurement of HBsAg and HBcAg expression in mice".
It seems that HBsAg is HBV surface antigen (HBsAg) and HBcAg is HBV core antigen (HBcAg). This should be indicated clearly.
- For figure 1D, line 273-276: "When qPCR was used to detect HBV particle secretion, HBV VPEs were detected in significant amounts after transduction with AAV2-A1 or AAV2-D3 (Figure 1d). These data demonstrate that the HBV genome-bearing AAVs successfully transduce liver-derived cells and lead to HBV gene expression and viral particle production."
With only this experiment in figure 1D, we still do not know whether these DNAs are from the HBV viral particle or AAV2-A1 and AAV2-D3 particles, as the qPCR primers used in this experiment can also amplify the original vectors AAV2-A1 and AAV2-D3, according figure 1A.
- Some spaces between words are too much, for examples in line 40:"A to I, and except for E and G," between for and E. Also line 43: "In South Africa where approximately 2.5 million", between In and South. These may be caused formatting or conversion, but can be easily improved.
Author Response
Dear Reviewer
Thank you very much for a thorough review of our manuscript and all comments are acknowledged. Attached is the response to the comments.
Regards
Betty

Round 2
Reviewer 1 Report
The author has addressed all issues, and thus the current manuscript can be accepted for publication.